# Accurate Layerwise Interpretable Competence Estimation

**Vickram Rajendran, William LeVine**
The Johns Hopkins University Applied Physics Laboratory
Laurel, MD 20723
`{vickram.rajendran, william.levine}@jhuapl.edu`

## Abstract

Estimating machine learning performance "in the wild" is both an important and unsolved problem. In this paper, we seek to examine, understand, and predict the pointwise competence of classification models. Our contributions are twofold: First, we establish a statistically rigorous definition of competence that generalizes the common notion of classifier confidence; second, we present the ALICE (Accurate Layerwise Interpretable Competence Estimation) Score, a pointwise competence estimator for any classifier. By considering *distributional, data,* and *model uncertainty*, ALICE empirically shows accurate competence estimation in common failure situations such as class-imbalanced datasets, out-of-distribution datasets, and poorly trained models.

Our contributions allow us to accurately predict the competence of any classification model given any input and error function. We compare our score with state-of-the-art confidence estimators such as model confidence and Trust Score, and show significant improvements in competence prediction over these methods on datasets such as DIGITS, CIFAR10, and CIFAR100.

## 1   Introduction

Machine learning algorithms have achieved tremendous success in areas such as classification [12], object detection [24], and segmentation [1]. However, as these algorithms become more prevalent in society it is essential to understand their limitations. In particular, a supervised machine learning model's performance on a reserved test point is characterized by the difference between that point's label and the model's prediction on that point. A model is considered performant on that point if this difference is sufficiently small; unfortunately, this difference is impossible to compute once the model is deployed since the point's true label is unknown.

This problem is exacerbated when we consider the difference between real world data and the curated datasets that the models are evaluated on — often these datasets are significantly different, and it is not clear whether performance on a held aside test set is indicative of real-world performance. It is essential to have a predictive measure of performance that does not require ground truth in order to determine whether or not a machine learning algorithm's prediction should be trusted "in the wild" — a measure of model competence. However, competence is currently not defined in any rigorous manner and is often restricted to the more specific idea of model confidence.

In this paper, we define competence to be a generalized form of predictive uncertainty, and so we must account for all of its' generating facets. Predictive uncertainty arises from three factors: *distributional, data*, and *model uncertainty*. *Distributional uncertainty* [4] arises from mismatched training and test distributions (i.e. dataset shift [23]). *Data uncertainty* [4] is inherent in the complex nature of the data (e.g. input noise, class overlap, etc.). Finally, *model uncertainty* measures error in the approximation of the true model used to generate the data (e.g. overfitting, underfitting, etc.) [4] —

this generally reduces as the amount of data increases. Accurate predictive uncertainty estimation (and thus accurate competence estimation) requires consideration of all three of these factors. Previous attempts to explicitly model these three factors require out-of-distribution data, or are not scalable to high dimensional datasets or deep networks [4] [18]; there are currently very few methods that do so in a way that requires no additional data, scales to high dimensional data and large models, and applies to any classification model, regardless of architecture, dataset, or performance.

We focus on mitigating these issues in the space of classifiers. In Section 2 we present several definitions, including a robust, generalizable definition of model *competence* that encompasses the common notion of model confidence. In Section 3 we examine the related work in the areas of predictive uncertainty estimation and interpretable machine learning. In Section 4 we show a general metric for evaluating competence estimators. In Section 5 we develop the "ALICE Score," an accurate layerwise interpretable competence estimator. In Section 6 we empirically evaluate the ALICE Score in situations involving different types of predictive uncertainty and on various models and datasets. We conclude in Section 7 with implications and ideas for future work.

## 2 Definitions

**Definition 1. (Error Function)** Let $\mathcal{C}$ be the finite "label space" of possible labels of the true model $f$ used to generate data, and let $\mathcal{Y}$ be the associated unit simplex of class probabilities, which we call the "distributional space". Let $\hat{\mathcal{Y}} \subseteq \mathcal{Y}$ be the space of possible outputs of a classifier $\hat{f}$ that approximates $f$. We will denote the classes in $\mathcal{C}$ predicted by these models (usually through an argmax) as $\hat{f}_c$ and $f_c$. An error function $\mathcal{E}$ is a function $\mathcal{E}\colon \mathcal{Y} \times \hat{\mathcal{Y}} \to \mathbb{R}^{\geq 0} \cup \{+\infty\}$, with the property that $\mathcal{E}(y, \hat{y}) = \infty$ when $y \in \overline{\hat{\mathcal{Y}}} \cap \mathcal{Y}$. This property intuitively means that the output of the error function is infinite if the true class is outside of the classifier's prediction space. Given a point $x$, we denote $\mathcal{E}(f(x), \hat{f}(x))$ the error of $\hat{f}$ on $x$. Common examples of error functions are mean squared error, cross-entropy error, and 0-1 error (the indicator that the classes predicted by $\hat{f}$ and $f$ are different). Note that an error function is distinct from a loss function since it is neither required to be differentiable nor continuous.

**Definition 2. (Confidence)** The commonly accepted definition of classifier confidence [19] [2] [18] [3] is the probability that the model's predicted class on an input $x$ is the true class of $x$. Explicitly, this is $p(f_c(x) = \hat{f}_c(x)|x, \hat{f})$. This is also the inverse of the **predictive uncertainty** [3] of a classifier, which is the probability that the model's prediction is incorrect [18].

While confidence is sufficient in many cases, we would like to have a more general and flexible definition that can be tuned towards a specific user's goals. For example, some users may be interested in top-$k$ error, cross-entropy or mean squared error instead of 0-1 error. We can model this by rewriting the confidence definition with respect to an error function $\mathcal{E}$:

$$p(f_c(x) = \hat{f}_c(x)|x, \hat{f}) = p(\mathcal{E}(f(x), \hat{f}(x)) = 0|x, \hat{f})$$

where $\mathcal{E}$ is the 0-1 error. We can now extend $\mathcal{E}$ beyond $\mathcal{E}_{0-1}$ to fit an end-user's goals. We can make this definition even more general by borrowing ideas from the Probable Approximately Correct (PAC) Learning framework [30] and allowing users to specify an error tolerance $\delta$. For example, some users may allow for their prediction error to be below a specific $\delta$ for their model to be considered competent. One could imagine that for highly precise problems with low threshold for error, $\delta$ would be quite low, while less stringent use-cases could allow for larger $\delta$'s. The relaxation of the prediction error leads to the generalized notion of $\delta$-**competence**, which we define as $p(\mathcal{E}(f(x), \hat{f}(x)) < \delta|x, \hat{f})$. Confidence can be recovered by setting $\mathcal{E} = \mathcal{E}_{\text{0-1}}$ and $\delta \in (0, 1)$.

Allowing both $\delta$ and $\mathcal{E}$ to vary gives fine control to an end-user about the details of a model's performance with respect to a specific error function.

**Definition 3. ($\delta$-$\epsilon$ Competence)** The *true $\delta$-competence* of a model at a given point is the binary variable $\mathcal{E}(f(x), \hat{f}(x)) < \delta|x, f, \hat{f}$ where $\mathcal{E}$ is an error function (Definition 1). Note that $\mathcal{E}$ becomes a random variable when $f$ is unknown since $\mathcal{E}$ is a deterministic function of the uncertain variable $f(x)$ — this notion of randomness is slightly distinct from treating $\hat{f}$ as a random variable due to finite data. Given that $f$ is unknown, we must estimate the $\delta$-**competence**, which can now be written as $p(\mathcal{E}(f(x), \hat{f}(x)) < \delta|x, \hat{f})$. Putting a risk threshold $\epsilon$ on the value of the $\delta$-competence leads us to the

following notion: A model is $\delta$-$\epsilon$ **competent** with respect to $\mathcal{E}$ at $x$ if $p(\mathcal{E}(f(x), \hat{f}(x)) < \delta | x, \hat{f}) > \epsilon$, or it is likely to be approximately correct.

This definition of competence allows a user to set a correctness threshold ($\delta$) on how close the prediction and the true output need to be in order to be considered approximately correct, as well as set a risk threshold ($\epsilon$) on the probability that this prediction is approximately correct with respect to any error function. These thresholds and error functions allow for a flexible definition of competence that can be adjusted depending on the application. This also follows the definition of trust in [14] as "the attitude that an agent will help achieve an individual's goals in a situation characterized by uncertainty and vulnerability."

Since we neither have access to labels nor have enough information to efficiently compute the true probability distribution $p(\mathcal{E}(f(x), \hat{f}(x)) < \delta | x, \hat{f})$ we seek to estimate this probability. We make this clear with the following definition:

**Definition 4. (Competence Estimator)** A competence estimator of a model $\hat{f}$ with respect to the error function $\mathcal{E}$ is a function $g_{\hat{f}} \colon \mathcal{X} \times \mathbb{R} \to [0, 1]$, where $\mathcal{X}$ is the space of inputs, that is a statistical point estimator of the true variable $\mathcal{E}(f(x), \hat{f}(x)) < \delta | x, \hat{f}, f$. In particular, $g_{\hat{f}}(x, \delta) = \hat{p}(\mathcal{E}(f(x), \hat{f}(x)) < \delta | x, \hat{f})$.

In the future we omit conditioning on $\hat{f}$ in our notation with the note that all subsequent probabilities are conditioned on $\hat{f}$.

# 3 Related Work

Competence estimation is closely tied with the well-studied areas of predictive uncertainty and confidence estimation, which can further be divided into Bayesian approaches such as [9] [7] [17], or non-Bayesian approaches including [5], [22], [13]. Bayesian methods attempt to determine some distribution about each of the weights in a network and predict a distribution of outputs using this distribution of weights. Computing the uncertainty of a prediction then becomes computing statistics about the estimated output distribution. These estimates tend to perform well, but tend not to be scalable to high dimensional datasets or larger networks. The non-bayesian methods traditionally fall under ensemble approaches [13], training on out-of-distribution data [18] [22] [29], or dropout [5]. This field tends to only work on a certain subset of classifiers (such as models with dropout for [5]) or require modifications to the models in order to compute uncertainty [19]. Many of these methods are based off of the unmodified model confidence [5], and thus could be supplementary to our new competence score. To the best of our knowledge there are no existing Bayesian or non-Bayesian methods that consider competence with respect to error functions other than 0-1 error nor methods that have tunable tolerance parameters.

Another related area of research is interpretable machine learning. Methods such as prototype networks [28] or LIME [25] are very useful in explaining why a classifier is making a prediction, and we expect these methods to augment our work. However, competence prediction does not attempt to explain the predictions of a classifier in any way—we simply seek to determine whether or not the classifier is competent on a point, without worrying about why or how the model made that decision. In this sense we are more closely aligned with *calibration* [6], which adjusts prediction scores to match class conditional probabilities which are interpretable scores [29] [31] and works such as [26] are orthogonal to ours. While our goal is not to compute class probabilites, our method similarly provides an interpretable probability score that the model is competent.

The closest estimators to our own are [2] and [8]. [2] learns a meta model that ensembles transfer classifiers' predictions to predict whether or not the overall network has a correct classification. Conversely, [8] computes the ratio of the distance to the predicted class and the second highest predicted class as a Trust Score. While [2] takes into account *data uncertainty* with transfer classifiers, it does not explicitly take into account *distributional* or *model uncertainty*. Oppositely, [8] considers neither *model* nor *data uncertainty* explicitly, though it does model *distributional uncertainty* similarly to [13], [15], and [16]. Further, both merely *rank* examples according to uncertainty measures that are not human-interpretable. They also focus on *confidence* rather than *competence*, which does not allow them to generalize to either more nuanced error functions or varying margins of error.

To the best of our knowledge, the ALICE Score is the first competence estimator that is scalable to large models and datasets and is generalizable to all classifiers, error functions, and performance levels. Our method takes into account all three aspects of predictive uncertainty in order to accurately predict competence on all of the models and datasets that it has encountered, regardless of the stage of training. Further, it does not require any out-of-distribution data to train on and can easily be interpreted as a probability of model competence. It also provides tunable parameters of $\delta$, $\epsilon$, and $\mathcal{E}$ allowing for a more flexible version of competence that can fit a variety of users' needs.

## 4 Evaluating Competence Estimators

### 4.1 Binary $\delta - \epsilon$ Competence Classification

We consider the task of pointwise binary competence classification. Given $f(x)$ and $\hat{f}(x)$, we can directly calculate $\mathcal{E}(f(x), \hat{f}(x))$ and thus the model's true $\delta$ competence on $x$. Given a competence estimator, we can then *predict* if the model is $\delta$ competent on $x$, thus creating a binary classification task parametrized by $\epsilon$. This allows us to use standard binary classification metrics such as Average Precision (AP) across all recall values to evaluate the competence estimator.

We note that the true model competence is nondecreasing as $\delta$ increases since we are strictly increasing the support. In particular, we have that the model is truly incompetent with respect to $\mathcal{E}$ on all points when $\delta = 0$, and the model is truly competent with respect to $\mathcal{E}$ on all points as $\delta \to \infty$ as long as $\mathcal{E}$ is bounded above. This makes it difficult to pick a single $\delta$ that is representative of the performance of the competence estimator on a range of $\delta$'s. To mitigate this issue we report *mean* AP over a range of $\delta$'s, as this averages the estimator's precision across these error tolerances.

Note that this metric only evaluates how well each estimator *orders* the test points based on competence, and does not consider the actual value of the score. We test this since some competence estimators (e.g. TrustScore) only seek to *rank* points based on competence and do not care what the magnitude of the final score is. As a technical detail, this means that we cannot parametrize the computation of Average Precision by $\epsilon$ (since some estimators don't output scores in the range [0, 1]), and must instead parametrize each estimator's AP computation separately by thresholding on that estimator's output.

## 5 The ALICE Score: $\delta - \epsilon$ competence estimation

We would like to determine whether or not the model is competent on a point without knowledge of ground truth, as in a test-set scenario where the user does not have access to the labels of a data point. Formally, given a $\delta$ and an input $x$, we want to estimate $p(\mathcal{E}(f(x), \hat{f}(x)) < \delta | x)$.

We write $p(\mathcal{E}(f(x), \hat{f}(x)) < \delta | x)$ as $p(\mathcal{E} < \delta | x)$, where $\mathcal{E}$ is the random variable that denotes the value of the $\mathcal{E}$ function given a point $x$ and its label $f(x)$. We begin by marginalizing over the possible label values $f(x) = c_j \in \mathcal{Y}$ (where $c_j$ is the one-hot label for class $j$):

$$p(\mathcal{E} < \delta | x) = \sum_{c_j \in \mathcal{Y}} p(\mathcal{E} < \delta | c_j, x) p(c_j | x) \tag{1}$$

$$= \sum_{c_j \in \hat{\mathcal{Y}} \cap \mathcal{Y}} p(\mathcal{E} < \delta | c_j, x) p(c_j | x) + \sum_{c_j \in \overline{\hat{\mathcal{Y}}} \cap \mathcal{Y}} p(\mathcal{E} < \delta | c_j, x) p(c_j | x) \tag{2}$$

$$= \sum_{c_j \in \hat{\mathcal{Y}}} p(\mathcal{E} < \delta | c_j, x) p(c_j | x) \tag{3}$$

Note that the $\mathcal{E}(c_j, \hat{f}(x))$ was defined to be $\infty$ when $c_j \in \overline{\hat{\mathcal{Y}}} \cap \mathcal{Y}$ (Definition 1), thus the rightmost summation in Equation 2 is 0 for all $\delta$. Furthermore, since $\hat{\mathcal{Y}} \subseteq \mathcal{Y}$ (Definition 1) we have $\hat{\mathcal{Y}} \cap \mathcal{Y} = \hat{\mathcal{Y}}$ which gives the final equality. To explicitly capture *distributional uncertainty*, we now marginalize over the variable $D$, which we define as the event that $x$ is in-distribution:

$$p(\mathcal{E} < \delta | x) = \sum_{c_j \in \hat{\mathcal{Y}}} p(\mathcal{E} < \delta | c_j, x) p(c_j | x)$$

$$= \sum_{c_j \in \hat{\mathcal{Y}}} p(\mathcal{E} < \delta | c_j, x, D) p(c_j | x, D) p(D | x) + \sum_{c_j \in \hat{\mathcal{Y}}} p(\mathcal{E} < \delta | c_j, x, \overline{D}) p(c_j | x, \overline{D}) p(\overline{D} | x)$$

$$(4)$$

Consider the rightmost summation in Equation 4. This represents the probability that the model is competent on the point $x$ assuming that $x$ is out-of-distribution. However, this term is intractable to approximate due to *distributional uncertainty*. Given only in-distribution training data, we assume that we cannot know whether the model will be competent on out-of-distribution test points. To mitigate this concern we lower bound the estimation by setting this term to 0 — this introduces the inductive bias that the model is not competent on points that are out-of-distribution. This simplification yields:

$$p(\mathcal{E} < \delta | x) \geq p(D | x) \sum_{c_j \in \hat{\mathcal{Y}}} p(\mathcal{E} < \delta | c_j, x) p(c_j | x, D) \qquad (5)$$

This allows our estimate to err on the side of caution as we would rather predict that the model is incompetent even if it is truly competent compared to the opposite situation. We approximate each of the terms in Equation 5 in turn.

## 5.1 Approximating $p(D|x)$

This term computes the probability that a point $x$ is in-distribution. We follow a method derived from the state-of-the-art anomaly detector [16] to compute this term: For each class $j$ we fit a class-conditional Gaussian $G_j$ to the set $\{x \in \mathcal{X}_{train} : \hat{f}(x) = c_j\}$ where $\mathcal{X}_{train}$ is the training data. Given a test point $x$ we then compute the Mahalanobis distance $d_j$ between $x$ and $G_j$. In order to turn this distance into a probability, we consider the empirical distribution $\beta_j$ of possible in-distribution distances by computing the distance of each training point to the Gaussian $G_j$, and then computing the survival function. We take the maximum value of the survival function across all $j$. This intuitively models the probability that the point is in-distribution with respect to *any* class. Explicitly, we have $p(D|x) = \max_j 1 - \text{CDF}_{\beta_j}(d_j)$. Note that this term measures distribution shift, which closely aligns with *distributional uncertainty*.

## 5.2 Approximating $p(\mathcal{E} < \delta | x, c_j)$

This term computes the probability that the error at the point $x$ is less than $\delta$ given that the one-hot label is $c_j$. We directly compute $\mathcal{E}(c_j, \hat{f}(x))$, then simply check whether or not this error is less than $\delta$. Note that this value is always 1 or 0 since it is the indicator $\mathbb{1}[\mathcal{E}(c_j, \hat{f}(x)) < \delta]$, and that this term estimates the difference between the predictions of $f$ and $\hat{f}$, which aligns with *model uncertainty*.

## 5.3 Approximating $p(c_j | x, D)$

This term computes the probability that a point $x$ is of class $j$, given that it is in-distribution. To estimate this class probability, we fit a transfer classifier at the given layer and use its class-probability output, $\hat{p}(c_j | x, D)$. Since the test points are assumed to be in-distribution, we can trust the output of the classifier as long as it is calibrated — that is, for all $x$ with $p(c_j | x) = p$, $p$ of them belong to class $j$. [21] examines the calibration of various classifiers, and shows that Logistic Regression (LR) Classifiers are well calibrated. Random Forests and Bagged Decision Trees are also calibrated [21], however, we find that the choice of calibrated classifier has little effect on the accuracy of our competence estimator. Note that — with a perfectly calibrated classifier — this term estimates the uncertainty inherent in the data (e.g. a red/blue classifier will always be uncertain on purple inputs due to class overlap), which closely aligns with *data uncertainty*.

## 5.4 The ALICE Score

Putting all of these approximations together yields the ALICE Score:

$$p(\mathcal{E}(f(x), \hat{f}(x)) < \delta | x) \gtrsim \max_j (1 - \mathrm{CDF}_{\beta_j}(d_j)) \sum_{c_j \in \hat{\mathcal{Y}}} \mathbb{1}[\mathcal{E}(\hat{f}(x), c_j) < \delta] \hat{p}(c_j | x, D) \quad (6)$$

Note that the ALICE Score can be written at layer $l$ of a neural network by treating $x$ as the activation of layer $l$ in a network and using those activations for the transfer classifiers and the class conditional Gaussians.

We do not claim that the individual components of the ALICE Score are optimal nor that our estimator is optimal — we merely wish to demonstrate that the ALICE framework of expressing competence estimation according to Equation 6 is empirically effective.

# 6 Experiments and Results

## 6.1 Experimental Setup

We conduct a variety of experiments to empirically evaluate ALICE as a competence estimator for classification tasks. We vary the model, training times, dataset, and error function to show the robustness of the ALICE Score to different variables. We compute metrics for competence prediction by simply using the score as a ranking and thresholding by recall values to compare with other scores that are neither $\epsilon$-aware nor calibrated, as discussed in Section 4. The mean Average Precision is computed across $100$ $\delta$'s linearly spaced between the minimum and maximum of the $\mathcal{E}$ output (e.g. for cross-entropy we space $\delta$'s between the minimum and the maximum cross-entropy error on a validation set). For all experiments, we compute ALICE scores on the penultimate layer, as we empirically found this layer to provide the best results — we believe this is due to the penultimate layer having the most well-formed representations before the final predictions. We compare our method only with Trust Score and model confidence (usually the softmax score) since they apply to all models and do not require extraneous data. Further experimental details are provided in Appendix A.

## 6.2 Predictive Uncertainty Experiments

Since competence is a generalized form of confidence, and confidence amalgamates all forms of predictive uncertainty, competence estimators must account for these factors as well. We empirically show that ALICE can accurately predict competence when encountering all three types of predictive uncertainty — note that we do not claim that the ALICE framework perfectly disentangles these three facets, merely that each term is essential to account for all forms of predictive uncertainty.

We first examine *model uncertainty* by performing an ablation study on both overfit and underfit classical models on DIGITS and VGG16 [27] on CIFAR100 [11]. Details about these models are in Appendix A. As expected, ALICE strongly outperforms the other metrics in areas of over and underfitting and weakly outperforms in regions where the network is trained well (Table 1). Further, we highlight a specific form of *model uncertainty* in Figure 1 by performing the same ablation study on the common situation of class-imbalanced datasets. We remove $95\%$ of the training data for the final 5 classes of CIFAR10 so that the model is poorly matched to these low-count classes, thus introducing model uncertainty. Figure 1 shows the mean Average Precision (mAP) of competence prediction on the unmodified CIFAR10 test set after fully training VGG16 on the class-imbalanced CIFAR10 dataset. While all of the metrics perform similarly on the classes of high count, neither softmax (orange) nor trust score (green) were able to accurately predict competence on the low count classes. ALICE (blue), on the other hand, correctly identifies competence on all classes because ALICE considers *model uncertainty*. We additionally show that omitting the term $p(\mathcal{E} < \delta | x, c_j)$ removes this capability, thus empirically showing that this term is necessary to perform accurate competence estimation under situations of model uncertainty.

While Figure 1 and Table 1 show ALICE's performance under situations of high *model uncertainty*, we show ALICE's performance under situations of *distributional uncertainty* in Table 2. First we

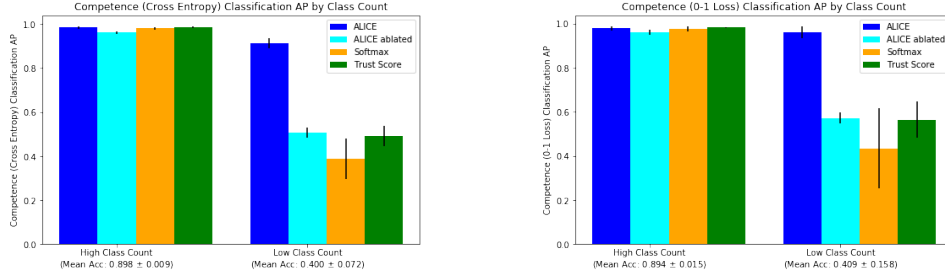

**(a)** mAP of competence scores ($\mathcal{E}$ = cross-entropy)　　　**(b)** mAP of competence scores ($\mathcal{E}$ = 0-1 error)

**Figure 1:** Competence Scores on Class Imbalanced CIFAR10

**Table 1:** mAP for Competence Prediction Under Model Uncertainty ($\mathcal{E}$ = cross-entropy). VGG16 is tested on CIFAR100 while the other models are on DIGITS. (U) is underfit, (W) is well trained, and (O) is overfit. Ablated ALICE refers to ALICE without the $p(\mathcal{E} < \delta | x, c_j)$ terms. Hyperparameters for these trials are in Appendix A.

| Model | Accuracy | Softmax | TrustScore | Ablated ALICE | ALICE |
|---|---|---|---|---|---|
| MLP (U) | .121 ± .048 | .0486 ± .015 | .505 ± .27 | .0538 ± .031 | **.999 ± .0015** |
| MLP (W) | .898 ±.022 | .989 ±.005 | .929 ±.044 | .958 ±.042 | **.998 ±.001** |
| MLP (O) | .097 ±.015 | .532 ±.062 | .768 ±.064 | .576 ±.033 | **.996 ±.003** |
| RF (U) | .563 ± .078 | .824 ± .16 | .504 ± .33 | .290 ± .322 | **.999 ± .0011** |
| RF (W) | .930 ±.019 | .998 ±.002 | .898 ±.025 | .923 ±.016 | **.999 ±.000** |
| SVM (U) | .630 ±.018 | .995 ±.003 | .626 ±.046 | .496 ±.069 | **1.00 ±.000** |
| SVM (W) | .984 ±.009 | **1.00 ±.000** | .931 ±.048 | .963 ±.038 | **1.00 ±.000** |
| SVM (O) | .258 ± .023 | .200 ± .16 | .215 ± .12 | .252 ± .16 | **.981 ± .028** |
| VGG16 (U) | .0878 ± .0076 | .899 ± .014 | .292 ± .049 | .0369 ± .0041 | **.913 ± .012** |
| VGG16 (W) | .498 ± .012 | .975 ± .013 | .604 ± .104 | .0863 ± .0071 | **.978 ± .0082** |
| VGG16 (O) | .282 ± .15 | .659 ± .024 | .665 ± .0080 | .257 ± .018 | **.738 ± .019** |

define a *distributional competence* error function:

$$\mathcal{E}_{\mathcal{D}}(f(x), \hat{f}(x)) = \begin{cases} 0 & f(x) \in \hat{\mathcal{Y}} \\ 1 & f(x) \notin \hat{\mathcal{Y}} \end{cases}$$

This function is simply an indicator as to whether or not the true label of a point is in the predicted label space. We fully train ResNet32 on the unmodified CIFAR10 training set. We then compute competence scores with respect to $\mathcal{E}_D$ on a test set with varying proportions of SVHN [20] (out-of-distribution) and CIFAR10 (in-distribution) data. In this case $\mathcal{Y} = \mathcal{Y}_{\text{CIFAR}} \cup \mathcal{Y}_{\text{SVHN}}$ but $\hat{\mathcal{Y}} = \mathcal{Y}_{\text{CIFAR}}$, thus $\mathcal{E}_D$ is 1 on SVHN points and 0 on CIFAR points. Table 2 shows that both softmax and ALICE without the $p(D|x)$ term perform poorly on distributional competence. In contrast, both the full ALICE score and Trust Score are able to estimate distributional competence in all levels of distributional uncertainty — this is expected since ALICE contains methods derived from a state-of-the-art anomaly detector [16] and Trust Score considers distance to the training data. Note that this construction of the distributional competence function is a clear example of how the general notion of competence can vary tremendously depending on the task at hand, and ALICE is capable of predicting accurate competence estimation for any of these notions of competence.

**Table 2:** mAP for Competence Prediction Under Distributional Uncertainty ($\mathcal{E} = \mathcal{E}_{\mathcal{D}}$).

| CIFAR/SVHN Proportion | Softmax | TrustScore | Ablated ALICE | ALICE |
|---|---|---|---|---|
| 10/90 | .458 ±0.056 | .518 ±0.039 | .100 ±0.000 | **.868 ±0.014** |
| 30/70 | .693 ±0.034 | .721 ±0.026 | .300 ±0.000 | **.946 ±0.007** |
| 50/50 | .816 ±0.020 | .833 ±0.015 | .500 ±0.000 | **.970 ±0.003** |
| 70/30 | .901 ±0.010 | .910 ±0.008 | .700 ±0.000 | **.985 ±0.002** |
| 90/10 | .970 ±0.003 | .972 ±0.002 | .900 ±0.000 | **.997 ±0.001** |

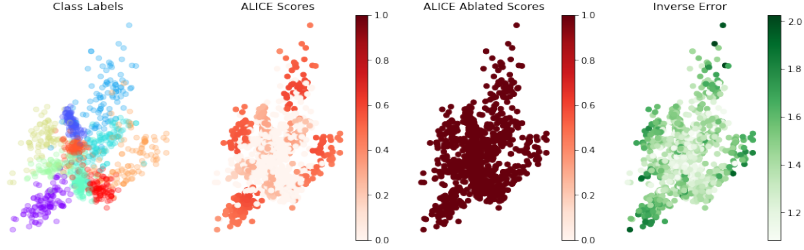

**Figure 2:** Competence Visualization on CIFAR10 ($\delta = .001$, $\mathcal{E} =$ cross-entropy). Points are projected to two dimensions with Neighborhood Component Analysis. From left to right, figures are colored by the class label, ALICE Score, Ablated ALICE Score, and inverse error (so darker colors imply competence).

We examine ALICE's capturing of *data uncertainty* by observing competence predictions in areas of class overlap in Figure 2. Here we trained VGG16 on CIFAR10 [10] and visualized competence scores with respect to cross-entropy. Note that the competence scores are very low in areas of class overlap, and that these regions also match with areas of high error. Additional experiments with varying models, error functions, and levels of uncertainty are provided in Appendix B.

### 6.3 Calibration Experiments

While the previous experiments show the ability of ALICE to *rank* points according to competence, we now show the interpretability of the ALICE score through calibration curves. Note that we are not attempting to intepret or explain *why* the model has made the decision that it has, we simply aim to show that the ALICE score matches its semantic meaning: for all points with ALICE score of $p$, we expect $p$ of them to be truly competent. To show this, we train ResNet32 on CIFAR100 and compute ALICE scores at various stages of training and for different error functions (we use $\delta = 0.2$ when computing competence for $\mathcal{E}_{\text{xent}}$. We bin the ALICE scores into tenths ([0.0 - 0.1], [0.1 - 0.2), ..., [0.9, 1.0)) and plot the true proportion of competent points for each bin as a histogram. Note that a perfect competence estimation with infinite data would result in these histograms roughly resembling a $y = x$ curve. We visualize the difference between our competence estimator and perfect competence estimation by showing these residuals as well as the number of points in each bin in Figure 3. Note that ALICE is relatively well-calibrated at all stages of training and for all error functions tested — this result shows that one can *interpret* the ALICE score as an automatically calibrated probability that the model is competent on a particular point. This shows that not only does the ALICE Score *rank* points accurately according to their competence but it also rightfully assigns the correct probability values for various error functions and at all stages of training.

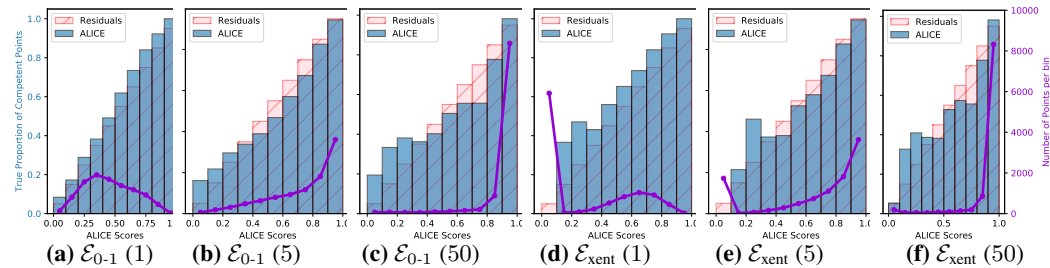

**Figure 3:** ALICE score calibration of ResNet32 trained on CIFAR10, with various error functions and stages of training. The captions show the error functions and number of epochs trained.

## 7 Conclusions and Future Work

In this work we present a new, flexible definition of competence. Our definition naturally generalizes the notion of confidence by allowing a variety of error functions as well as risk and correctness thresholds in order to construct a definition that is tunable to an end-user's needs. We also develop the ALICE Score, an accurate layerwise interpretable competence estimator for classifiers. The

ALICE Score is not only applicable to any classifier but also outperforms the state-of-the-art in competence prediction. Further, we show that the ALICE Score is robust to out-of-distribution data, class imbalance and poorly trained models due to our considerations of all three facets of predictive uncertainty.

The implications of an accurate competence estimator are far reaching. For instance, future work could include using the ALICE Score to inform an Active Learning acquisition function by labeling points that a model is least competent on. One could also examine a network more closely by performing feature visualization or finding prototypes in areas of low competence, as this would elucidate which features are correlated with incompetence. This is particularly useful since the ALICE Score can be computed layerwise in order to find both low and high level features that the model is not competent on. Competence estimators could also be used as test and evaluation metrics when a model is deployed to detect both distributional shift and classification failure.

Future work will focus on extending the ALICE Score to supervised tasks other than classification such as object detection, segmentation, and regression. Additionally, because many of the components of the ALICE Score are state-of-the-art for detecting adversarial examples, we expect that the ALICE Score would also be able to detect adversarial samples and assign them low competence, though we have not tested this explicitly. Further research will also include better approximations of the terms in the ALICE Score to improve competence estimation. Finally, we plan to explore different methods to ensemble the layerwise ALICE Scores into an overall ALICE Score for the model and determine whether or not that improves performance compared to the layerwise ALICE Scores.

**Acknowledgements**

The authors would like to thank the JHU/APL Internal Research and Development (IRAD) program for funding this research.

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
