[Supplementary Material · ALICE_CAMERA_SUPP.pdf]

# 8 Appendix A: Experimental Details

For the CIFAR experiments we initialize VGG16 with pretrained imagenet weights and for the DIGITS experiments we use classical machine learning models such as Support Vector Machines (SVMs), Random Forests (RFs), Multilayer Perceptrons (MLPs), and Logistic Regression (LR). VGG16 and LeNET are trained with learning rates of $1e - 4$ and $3e - 4$ respectively, and default regularization parameters. The deep models are trained on a single NVIDIA 1080TI GPU, while the classical models are all trained on the CPU. For all experiments we perform ten trials and report the mean and standard deviation. The default scikit-learn parameters are used for SVM's, LR, RF, MLP unless stated otherwise with the exception of the SVM regularization parameter $C$ which we set to 0.01. We use an MLP with one hidden layer of 10 neurons. The grid search for ALICE's transfer classifier regularization parameter is over a logspace from $-5$ to $5$.

For the model uncertainty experiments we have trained the MLP for 1 iteration to ensure underfitting. We set the RF to have max depth = 1 and 20 estimators to also ensure the model will poorly match the data. The SVM has a degree 5 polynomial kernel and no regularization to ensure overfitting to the data. The well-trained VGG16 network has a regularization parameter of 1e+8 on the final two layers and is fully trained. The underfit and overfit networks both have a regularization parameter of 0, but are trained for 1 and 100 epochs respectively.

Each dataset is randomly split (unless a split already exists) into an $80 - 10 - 10$ split of train-test-validation, and each model is trained until the performance on a validation set is maximized unless explicitly stated otherwise. The deep models are all trained with respect to cross-entropy loss, while the classical models are fitted with their respective loss functions.

# 9 Appendix B: Further Competence Prediction Experiments

Here we show competence prediction on various models with respect to different error functions on the DIGITS and CIFAR100 datasets. Note that ALICE consistently performs well regardless of model type or model accuracy, and significantly outperforms both Trust Score and model confidence on almost every model except for SVM with a polynomial kernel, where it loses by a statistically insignificant amount. Note that as the model accuracy increases, the improvement of ALICE over the other methods decreases — this is because model uncertainty is decreasing, which alleviates the failures of methods that do not consider model uncertainty.

## 9.1 Competence Prediction (Cross-Entropy)

**Table 3:** mAP for Competence Prediction on DIGITS ($\mathcal{E}$ = cross-entropy)

| Model | Accuracy | Softmax | TrustScore | ALICE |
|---|---|---|---|---|
| SVM (RBF) | .147 ± .032 | .414 ± .15 | .346 ±.086 | **.989 ± .0078** |
| SVM (Poly) | .988 ± .007 | **1.00 ± .00027** | .949 ± .0073 | .999 ± 0.0011 |
| SVM (Linear) | .971 ± .011 | **.999 ± .00066** | .951 ± .0094 | **.999 ± .00084** |
| RF | .928 ± .013 | .998 ± .0014 | .876 ± .013 | **1.00 ± .00048** |
| MLP (5 Iterations) | .158 ± .056 | .217 ± .13 | .579 ± .12 | **.966 ± .047** |
| MLP (200 Iterations) | .925 ± .017 | .988 ± .0087 | .963 ± .014 | **.998 ± .0017** |
| LR | .946 ± .017 | .995 ± .0029 | .977 ± .0041 | **.998 ± .0015** |

**Table 4:** mAP for Competence Prediction on CIFAR100 ($\mathcal{E}$ = cross-entropy)

| Model | Accuracy | Softmax | Trust Score | ALICE |
|---|---|---|---|---|
| ResNet50 (1 epoch) | .074 | .537 | .346 | **.723** |
| ResNet50 (5 epochs) | .293 | .895 | .715 | **.925** |
| ResNet50 (30 epochs) | .421 | .766 | .776 | **.828** |
| ResNet50 (100 epochs) | .450 | .795 | .807 | **.809** |

## 9.2 Competence Prediction (MSE)

**Table 5:** mAP for Competence Prediction on DIGITS ($\mathcal{E}$ = mean-squared-error)

| Model | Accuracy | Softmax | TrustScore | ALICE |
| --- | --- | --- | --- | --- |
| SVM (RBF) | .147 ± .032 | .394 ± .066 | .361 ± .046 | **.985 ± .011** |
| SVM (Poly) | .988 ± .007 | .999 ± .0018 | .990 ± .0045 | **.998 ± 0.0012** |
| SVM (Linear) | .971 ± .011 | 1.00 ± .00065 | .994 ± .0037 | **.999 ± .0013** |
| RF | .928 ± .013 | .996 ± .0016 | .956 ± .012 | **.999 ± .00034** |
| MLP (5 Iterations) | .158 ± .056 | .384 ± .11 | .746 ± .049 | **.992 ± .015** |
| MLP (200 Iterations) | .925 ± .017 | .986 ± .0069 | .985 ± .012 | **.997 ± .0027** |
| LR | .946 ± .017 | .995 ± .0025 | .989 ± .0051 | **.998 ± .0015** |

## 9.3 Competence Prediction ($\mathcal{E}$ = 0-1 error)

**Table 6:** AP for Competence Prediction on DIGITS ($\mathcal{E}$ = 0-1 error)

| Model | Accuracy | Softmax | TrustScore | ALICE |
| --- | --- | --- | --- | --- |
| SVM (RBF) | .147 ± .032 | .142 ± .27 | .106 ± .038 | **.983 ± .020** |
| SVM (Poly) | .988 ± .007 | **.999 ± .0013** | **.999 ± .00057** | **.999 ± .00042** |
| RF | .928 ± .013 | .994 ± .0034 | **1.00 ± .00050** | .998 ± .0015 |
| SVM (Linear) | .971 ± .011 | **.999 ± .0011** | **.999 ± .00091** | .997 ± .0012 |
| MLP (5 Iterations) | .158 ± .056 | .178 ± .094 | **.996 ± .0036** | .984 ± .014 |
| MLP (200 Iterations) | .925 ± .017 | .981 ± .014 | **.999 ± .00037** | **.999 ± .0014** |
| LR | .946 ± .017 | .994 ± .0027 | **.999 ± .00038** | .996 ± .0017 |

**Table 7:** AP for Competence Prediction on CIFAR100 ($\mathcal{E}$ = 0-1 error)

| Model | Accuracy | Softmax | TrustScore | ALICE |
| --- | --- | --- | --- | --- |
| VGG16 (1 epoch) | .209 | .514 | .654 | **.696** |
| VGG16 (5 epochs) | .323 | .670 | **.762** | .756 |
| VGG16 (30 epochs) | .513 | .845 | .867 | **.873** |
| VGG16 (100 epochs) | .536 | .803 | .863 | **.871** |

## 9.4 Histogram of ALICE Scores on in and out-of-distribution data

**(a)** Competence scores on MNIST test set

**(b)** Competence scores on CIFAR100 test set

**Figure 4:** Competence Scores ($\mathcal{E} = \mathcal{E}_{\mathcal{D}}$) on In-Distribution (MNIST) and Out-of-Distribution (CIFAR100) Data for LeNet trained on MNIST. ALICE Ablated refers to ALICE without the $p(D|x)$ term. Note how both the ALICE Scores and the ablated ALICE scores are both very high on the in-distribution examples; however, only the unablated ALICE scores are rightfully low when the model sees images from CIFAR100.

**Table 8:** mAP for Competence Prediction Under Class Overlap ($\mathcal{E} = \mathcal{E}_{0-1}$). The datasets are synthetic datasets designed to show class overlap. Given a parameter $z$, we construct the dataset $D_z$ as follows. Class 0 is a uniform distribution $U(-5, z)$, and class 1 is a uniform distribution $U(-z, 5)$. Shifting $z$ from 0 to 5 yields different class overlap percentages. For the training set, we randomly generate 1000 points from each class distribution; for the test and validation sets we randomly generate 100 points from each class. We train a Logistic Regression Model and compute competence prediction scores with $\mathcal{E} = \mathcal{E}_{0-1}$. Ablated ALICE refers to ALICE without the $p(c_j|x, D)$ term. In contrast to the full ALICE score, ALICE ablated is unable to accurately predict competence in situations of class overlap. Additionally note that when there is 100 percent class overlap the model is randomly guessing, thus pointwise competence is also random.

| Overlap Percentage | Accuracy | Softmax | TrustScore | Ablated ALICE | ALICE |
|---|---|---|---|---|---|
| 0.00 | $1.00 \pm 0.0$ | $\mathbf{1.00 \pm 0.0}$ | $\mathbf{1.00 \pm 0.0}$ | $\mathbf{1.00 \pm 0.0}$ | $\mathbf{1.00 \pm 0.0}$ |
| 0.10 | $.945 \pm 0.0$ | $\mathbf{.998 \pm .000021}$ | $\mathbf{.998 \pm 0.0}$ | $.945 \pm 0.0$ | $\mathbf{.998 \pm 0.0}$ |
| 0.25 | $.865 \pm 0.0$ | $.986 \pm .000015$ | $982 \pm 0.0$ | $.865 \pm 0.0$ | $\mathbf{.987 \pm 0.0}$ |
| 0.50 | $.730 \pm 0.0$ | $\mathbf{.960 \pm .000020}$ | $.948 \pm 0.0$ | $.730 \pm 0.0$ | $\mathbf{.960 \pm 0.0}$ |
| 0.75 | $.625 \pm 0.0$ | $\mathbf{.862 \pm 0.0}$ | $.823 \pm 0.0$ | $.625 \pm 0.0$ | $\mathbf{.862 \pm 0.0}$ |
| 1.00 | $.535 \pm 0.0$ | $.499 \pm 0.0$ | $.530 \pm 0.0$ | $\mathbf{.535 \pm 0.0}$ | $.500 \pm 0.0$ |

**Figure 5:** $\delta$ vs mean ALICE Score across all points on MNIST ($\mathcal{E}$ = cross-entropy).

**Figure 6:** $\delta$ vs mean ALICE Score across all points on CIFAR10 ($\mathcal{E}$ = cross-entropy).

# 10    Appendix C: Competence Visualization

**(a)** $\delta = 0$

**(b)** $\delta = 1e - 3$

**(c)** $\delta = 1e - 1$

**(d)** $\delta = 2e - 1$

**Figure 7:** Competence Scores at varying $\delta$'s on the CIFAR10 Dataset with $\mathcal{E}$ =cross-entropy. Points were projected to two dimensions with Neighborhood Component Analysis. The left images are colored based on the actual class label. The middle images are colored based on the predicted $\delta$-competence at that specific $\delta$. ALICE Ablated refers to ALICE with $p(c_j|x_i, D)$ removed. Finally, the right images show the inverse pointwise true error. Note how points increase in their ALICE Score as the error threshold increases. When $\delta = 0.2$ (the max value of $\mathcal{E}$ in the val set), the model is rightfully considered competent on nearly all points, and when $\delta = 0$ (the absolute min value of $\mathcal{E}$) the model is considered incompetent on nearly all points, as desired.