[Reviews · NeurIPS 2019]

Reviewer 1



Summary ======= The authors propose to adapt the common definition of a model's *predictive confidence* in the sense of the PAC framework. They introduce a formal definition of the mentioned adaption of predictive confidence, which generalizes classical confidence (1) by a notion of (meta) uncertainty and (2) to arbitrary error functions beyond the 0-1-Error. It is proposed that predictive uncertainty is generated by 3 main factors, i.e., distributional, data, and model uncertainty. Under this assumption a uncertainty model is introduced by formulating one choice of estimation for each of those 3 factors. In an empiric study the authors aim at showing that (1) the proposed estimator outperforms current state of the art and (2) their factorization (distributional, data, and model uncertainty) is indeed close to the true situation. Pros ==== 1. The authors tackle an indeed important topic which needs further attention. 2. The general idea of integrating measures of predictive uncertainty into a PAC way of thinking seem promising. 3. It seems reasonable to factorize predictive uncertainty and address/estimate the factors independently and the presented experiments (partly) support the proposed factorization. Cons ==== 1. Although interesting, it is hard to follow the authors particular choice of predictive uncertainty factorization; the wording is suggesting that this choice is obvious but little motivation is given. For example, the distinction between distributional and data uncertainty seems to be rather fuzzy and view point dependent. 2. One claimed key contribution of this work is a rigorous generalization of confidence to the PAC frame work and to arbitrary error functions. With this in mind, its introduction (Sec. 2) lacks mathematical precision. Some concerns: 2.1 There are implicit assumptions on Y, e.g., countability. If the authors claim to introduce a rigorous definition there must not be implicit assumptions. 2.2 In Eq. 0 (the unnumbered before Eq. 1) there seems to be an algebra defined on Y as the term f(x) - \hat{f}(x) is assumed to be defined. This puzzles me as from the wording Y seems to be some sort of countable label space. 2.3 Eq. 1 seems to capture a triviality. What is the sense of this transformation? From my perspective the common root of most of those issues is that the authors formally do not distinguish between the label space and the distributional space and use them interchangeably. To be precise, for a finite label space the corresponding predictions, yielded by the model, are points on the unit simplex (here an algebra is defined). The corresponding label prediction is commonly the argmax of the label-wise probabilities. 3. I do not understand how the proposed score is evaluated. In Sec. 4 this is introduced but the last paragraph is not understandable to me. It seems that first the obvious way for evaluation is introduced (Paragraph 1+2) but then this is somehow changed to address the shortcomings of the competitors, but it is not clear to me how this is changed. Maybe a formula would help here. In this situation I cannot interpret the experimental results. 4. The selection of models in Table 1 puzzles me. For each model there should be a U, O, and W version. Why are some of those omitted? 5. The experimental setup addressing *distributional uncertainty* seems a little insidious. Given that MNIST is gray-valued and CIFAR100 is RGB it seems not difficult to succeed here. This is also reflected in the experimental outcome which is alarmingly good and indicating that the experimental design is probably not appropriate. Alternatively one could test CIFAR100 vs CIFAR10.

Reviewer 2



1. Originality The novel formulation of model competence is conceptually simple and straightforward, yet original. Although the proposed method depends largely on the state-of-the-art anomaly detector (Lee et al. 2018) for modeling distributional uncertainty and the simple transfer classifiers for data uncertainty, the novel formulation of delta competence enables simple yet effective modeling of model uncertainty for any classifier with any error function. As a combination, the proposed method is original as well as conceptually appealing. 2. Clarity This paper is very well-organized and clearly written. Introduction and related work are sufficient for non-experts to easily absorb the background, motivation, and significance of the problem of interest. Mathematical equations and their derivations are very well complemented by verbal descriptions, making the paper highly readable and understandable. 3. Quality The quality of the paper is acceptable. Although the predictive uncertainty experiments (Sec. 6.3) were somewhat rudimentary and limited to proof-of-concept (showing the robustness of the ALICE score in several manipulated/ablated scenarios), the authors explicitly acknowledged this limitation and were careful enough about their claims (L206-208 and L223-226). However, the calibration experiments (Sec. 6.3) were unsatisfactory. Interpretability is illustrated as an important aspect of the proposed method as suggested by its name (AL*I*CE), but it is rather misleading to claim that the ALICE score is well-calibrated and interpretable. Although L114-119 properly add caveats, it was not adequately explained what “an interpretable probability score with inherent meaning” (L119) actually means. Both terms “calibration” and “interpretability” were overgenerously exploited. The calibration of the ALICE score against delta is not what most readers would expect from the term “calibration,” and the claimed interpretability is quite distant from what most readers would expect from the term “human interpretable.” In order for the ALICE score to be maximally practically effective, it has to be well-calibrated (or at least calibratable) against class conditional probabilities (e.g. Fig. 1 of Guo et al. 2017). 4. Significance This paper seems to be fairly significant. I would expect the present work could be a good reference point for other researchers to further develop the core ideas presented in the paper, such as considering all aspects of predictive uncertainty (distributional, model, and data) and developing practical methods with generality and scalability. Arguably, simplicity of the proposed method and exceptional clarity of the paper add additional significance to this paper, increasing the chance that this paper is widely read and being motivational. Minor comments - Why is the maximum value of the cross-entropy error function 0.2 (L217 and Fig. 3)? - In Fig. 2, it is unclear what exactly “inverse true error” means. - Was there *any* previous attempt to consider all aspects of predictive uncertainty (distributional, model, and data uncertainty)? L39-41 is unclear whether there exists *any* line of research.

Reviewer 3



I believe that the core algorithmic idea that is introduced in this paper makes sense, and the results indicate that the method could be useful to evaluate confidence of classification models. I have some reservation however with respect to the quality of presentation, and I am not convinced that the paper is ready for publication in this form. I don't agree that mentioning PAC learning is necessary. The authors attach a name (i.e., model competence) to a particular, approximate confidence interval. If the authors want to refer to PAC learning, they should have some bounds that would include the sample size. This is missing in the paper, and for this reason I don't think that these methods should be put in the context of PAC learning. Line 26 says "it is not clear whether performance on a held aside test set is indicative of real-world performance ...", but the authors don't address this problem in the paper. It seems that the introduction does not match the subsequent sections. Specifically, the paragraph that is above Eq. 6 says "we cannot know whether the model will be competent on out-of-distribution test points". The arguments and statements are not coherent. There is something wrong with Eq. 1. I don't think that line 2 should start with "=". The explanation of randomness in Definition 3 is not clear to me. Isn't it the case that randomness comes from the limited training data, and every time we sample a new training dataset, we can get a slightly different set of data points? Also, I don't think that we can say that f is a variable. f is a function. Line 116 says "we simply seek to determine whether or not the classifier is competent on a point.". The authors should be careful with such statements because researchers who study interpretability would disagree with such views. See this paper: Stop Explaining Black Box Machine Learning Models for High Stakes Decisions and use Interpretable Models Instead, Nature Machine Intelligence. Cynthia Rudin "pointwise binary competence classification" should be explained before it is used in line 138 "pointwise rankings" in line 150 should be explained. Most of the section 5 looks good to me, and it presents and interesting heuristic approach that leads to promising results. However, as long as section 5.1 clearly describes the method, the method proposed by [16] evaluates the property of the dataset. So, this method could potentially be used to evaluate any model because it evaluates the data, i.e., it estimates the probability that a data point is in-distribution. This evaluation does not depend on any model, and all models can equally benefit from the potential of this method. Section 5.2 is not clear to me. The authors say "we directly compute \mathcal{E}...". Does it mean that reclassification accuracy is used here? If so, on which data, training, validation or testing? What are the "transfer classifiers" in line 204? It is very suspicious that the ALICE scores are used on the penultimate layers. This design choice should be carefully explained using theoretical arguments or empirical evidence. What is the role of softmax in line 246? What is it used for? It's use should be explained.

[Author Response · NeurIPS 2019]

The authors would like to thank the reviewers for their thoughtful comments. Our responses are below:

We have replaced the calibration experiment. In this new experiment, we aim to show that the ALICE score matches
its semantic meaning: for all points with ALICE score of $p$, we expect $p$ of them to be truly competent. To show this,
we bin the ALICE scores into tenths ([0.0 - 0.1), [0.1 - 0.2), ..., [0.9, 1.0)) and plot the true proportion of competent
points for each bin as a histogram. Note that a perfect competence estimation would result in these histograms roughly
resembling a $y = x$ curve. We visualize the difference between our competence estimator and perfect competence
estimation by showing these residuals as well as the number of points in each bin in Figure 1. Note that ALICE is
relatively well-calibrated at all stages of training and for all error functions tested. We would like to make clear that all
mentions of the word *interpretable* refer to this interpretability of the ALICE score — *not* the interpretability of any
machine learning model's predictions, as we state in Line 115.

**Figure 1:** ALICE score calibration of ResNet32 trained on CIFAR10. At 50 epochs we reach max validation accuracy. Full experimental details are in the final version.

(a) $\mathcal{E}_{0\text{-}1}$, epochs = 1    (b) $\mathcal{E}_{\text{xent}}$, epochs = 1    (c) $\mathcal{E}_{0\text{-}1}$, epochs = 5    (d) $\mathcal{E}_{\text{xent}}$, epochs = 5    (e) $\mathcal{E}_{0\text{-}1}$, epochs = 50    (f) $\mathcal{E}_{\text{xent}}$, epochs = 50

We have replaced the last paragraph of section 4 for futher clarity. It now reads: "Note that this metric only evaluates
how well each estimator *orders* the test points based on competence, and does not consider the actual value of the score.
We test this since some competence estimators (e.g. TrustScore) only seek to *rank* points based on competence and
do not care what the magnitude of the final score is. As a technical detail, this means that we cannot parametrize the
computation of Average Precision by $\epsilon$ (since some estimators don't output scores in the range [0, 1]), and must instead
parametrize each estimator's AP computation separately by thresholding on that estimator's output."

We have redone the distributional uncertainty experiment to follow standard out-of-distribution (OOD) detection
experiments. We train ResNet32 on CIFAR10 (in-distribution) and estimate the distributional competence (Line 241)
of images from SVHN (OOD). The results are below in Table 1. We have also revised Table 1 in the paper to have
underfit, wellfit, and overfit models for each model. The missing results are denoted below in Table 2. We have omitted
RF (O) since our random forest did not overfit. Further experimental details will be in the camera-ready version.

**Table 1:** mAP for Distributional Uncertainty ($\mathcal{E} = \mathcal{E}_{\mathcal{D}}$).

| Split | Softmax | TrustScore | Abl. ALICE | ALICE |
|---|---|---|---|---|
| 10/90 | .458 ±.056 | .518 ±.039 | .100 ±0.000 | **.868 ±0.014** |
| 30/70 | .693 ±0.034 | .721 ±0.026 | .300 ±0.000 | **.946 ±0.007** |
| 50/50 | .816 ±0.020 | .833 ±0.015 | .500 ±0.000 | **.970 ±0.003** |
| 70/30 | .901 ±0.010 | .910 ±0.008 | .700 ±0.000 | **.985 ±0.002** |
| 90/10 | .970 ±0.003 | .972 ±0.002 | .900 ±0.000 | **.997 ±0.001** |

**Table 2:** Continuation of Table 1 from the paper. ($\mathcal{E} = \mathcal{E}_{\text{xent}}$).

| Model | Softmax | TrustScore | Abl. ALICE | ALICE |
|---|---|---|---|---|
| MLP (W) | .989 ±.005 | .929 ±.044 | .958 ±.042 | **.998 ±.001** |
| MLP (O) | .532 ±.062 | .768 ±.064 | .576 ±.033 | **.996 ±.003** |
| RF (W) | .998 ±.002 | .898 ±.025 | .923 ±.016 | **.999 ±.000** |
| SVM (U) | .995 ±.003 | .626 ±.046 | .496 ±.069 | **1.00 ±.000** |
| SVM (W) | **1.00 ±.000** | .931 ±.048 | .963 ±.038 | **1.00 ±.000** |

We have modified all mentions of PAC Learning to clarify that our method is inspired, not derived, from PAC methods.

We have added to lines 39-41 to articulate prior work and motivate our usage of the three types of uncertainty and
the limitations of these works. It now reads: "Previous attempts to explicitly model these three factors require
out-of-distribution data, or are not scalable to high dimensional datasets or deep networks [1]".

We have edited the notation to distinguish between a finite "label space" $\mathcal{C}$ and the associated unit simplex, or
"distributional space" $\mathcal{Y}$, and have revised parts of the paper to accomodate (e.g. Eq. 0, the unnumbered before Eq. 1).

On Line 70, we remove Eq. 1 and clarify: "The relaxation of the prediction error leads to the generalized notion of
$\delta$-**competence**, which we define as $p(\mathcal{E} < \delta | x, \hat{f})$. Confidence can be recovered by setting $\mathcal{E} = \mathcal{E}_{0\text{-}1}$ and $\delta \in (0, 1)$."

We have revised Line 219 to specifically link model confidence and softmax together so that the role of softmax in line
246 is clear (another score that does not require ground truth to compute and can be treated as a competence estimator).

Several other minor comments (clarifying definitions e.g. "pointwise rankings," "inverse true error," "randomness" etc.,
clarifying ambiguity between calibration and interpretability) have been assimilated.

## Footnotes

[1] Malinin, Andrey, and Mark Gales. "Predictive uncertainty estimation via prior networks." Advances in Neural Information Processing Systems. 2018.; Yarin Gal, "Uncertainty in Deep Learning", Ph.D. thesis, University of Cambridge, 2016; Kimin Lee, Honglak Lee, Kibok Lee, and Jinwoo Shin, "Training confidence-calibrated classifiers for detecting out-of-distribution samples," International Conference on Learning Representations, 2018.


[Meta-Review · NeurIPS 2019]

This paper proposes interesting direction of defining competence score. While we view the mention of PAC learning unnecessary, there are interesting ideas in this paper worth sharing with the community. Please take extra care for R3's thoughtful and detailed comments on revision - these are concrete suggestions to improve your paper.